# Spherical Random Features for Polynomial Kernels

Jeffrey Pennington        Felix X. Yu        Sanjiv Kumar

Google Research
{jpennin, felixyu, sanjivk}@google.com

## Abstract

Compact explicit feature maps provide a practical framework to scale kernel methods to large-scale learning, but deriving such maps for many types of kernels remains a challenging open problem. Among the commonly used kernels for non-linear classification are polynomial kernels, for which low approximation error has thus far necessitated explicit feature maps of large dimensionality, especially for higher-order polynomials. Meanwhile, because polynomial kernels are unbounded, they are frequently applied to data that has been normalized to unit $\ell_2$ norm. The question we address in this work is: if we know *a priori* that data is normalized, can we devise a more compact map? We show that a putative affirmative answer to this question based on Random Fourier Features is impossible in this setting, and introduce a new approximation paradigm, Spherical Random Fourier (SRF) features, which circumvents these issues and delivers a compact approximation to polynomial kernels for data on the unit sphere. Compared to prior work, SRF features are less rank-deficient, more compact, and achieve better kernel approximation, especially for higher-order polynomials. The resulting predictions have lower variance and typically yield better classification accuracy.

## 1   Introduction

Kernel methods such as nonlinear support vector machines (SVMs) [1] provide a powerful framework for nonlinear learning, but they often come with significant computational cost. Their training complexity varies from $O(n^2)$ to $O(n^3)$, which becomes prohibitive when the number of training examples, $n$, grows to the millions. Testing also tends to be slow, with an $O(nd)$ complexity for $d$-dimensional vectors.

Explicit kernel maps provide a practical alternative for large-scale applications since they rely on properties of linear methods, which can be trained in $O(n)$ time [2, 3, 4] and applied in $O(d)$ time, independent of $n$. The idea is to determine an explicit nonlinear map $Z(\cdot) : \mathbb{R}^d \to \mathbb{R}^D$ such that $K(\mathbf{x}, \mathbf{y}) \approx \langle Z(\mathbf{x}), Z(\mathbf{y}) \rangle$, and to perform linear learning in the resulting feature space. This procedure can utilize the fast training and testing of linear methods while still preserving much of the expressive power of the nonlinear methods.

Following this reasoning, Rahimi and Recht [5] proposed a procedure for generating such a non-linear map, derived from the Monte Carlo integration of an inverse Fourier transform arising from Bochner's theorem [6]. Explicit nonlinear random feature maps have also been proposed for other types of kernels, such as intersection kernels [7], generalized RBF kernels [8], skewed multiplicative histogram kernels [9], additive kernels [10], and semigroup kernels [11].

Another type of kernel that is used widely in many application domains is the polynomial kernel [12, 13], defined by $K(\mathbf{x}, \mathbf{y}) = (\langle \mathbf{x}, \mathbf{y} \rangle + q)^p$, where $q$ is the bias and $p$ is the degree of the polynomial. Approximating polynomial kernels with explicit nonlinear maps is a challenging problem, but substantial progress has been made in this area recently. Kar and Karnick [14] catalyzed this line of

research by introducing the Random Maclaurin (RM) technique, which approximates $\langle \mathbf{x}, \mathbf{y} \rangle^p$ by the product $\prod_{i=1}^{p} \langle \mathbf{w}_i, \mathbf{x} \rangle \prod_{i=1}^{p} \langle \mathbf{w}_i, \mathbf{y} \rangle$, where $\mathbf{w}_i$ is a vector consisting of Bernoulli random variables. Another technique, Tensor Sketch [15], offers further improvement by instead writing $\langle \mathbf{x}, \mathbf{y} \rangle^p$ as $\langle \mathbf{x}^{(p)}, \mathbf{y}^{(p)} \rangle$, where $\mathbf{x}^{(p)}$ is the $p$-level tensor product of $\mathbf{x}$, and then estimating this tensor product with a convolution of count sketches.

Although these methods are applicable to any real-valued input data, in practice polynomial kernels are commonly used on $\ell_2$-normalized input data [15] because they are otherwise unbounded. Moreover, much of the theoretical analysis developed in former work is based on normalized vectors [16], and it has been shown that utilizing norm information improves the estimates of random projections [17]. Therefore, a natural question to ask is, *if we know* a priori *that data is $\ell_2$-normalized, can we come up with a better nonlinear map?*[1] Answering this question is the main focus of this work and will lead us to the development of a new form of kernel approximation.

Restricting the input domain to the unit sphere implies that $\langle \mathbf{x}, \mathbf{y} \rangle = 2 - 2||\mathbf{x} - \mathbf{y}||^2$, $\forall x, y \in \mathcal{S}^{d-1}$, so that a polynomial kernel can be viewed as a shift-invariant kernel in this restricted domain. As such, one might expect the random feature maps developed in [5] to be applicable in this case. Unfortunately, this expectation turns out to be false because Bochner's theorem cannot be applied in this setting. The obstruction is an inherent limitation of polynomial kernels and is examined extensively in Section 3.1. In Section 3.2, we propose an alternative formulation that overcomes these limitations by approximating the Fourier transform of the kernel function as the positive projection of an indefinite combination of Gaussians. We provide a bound on the approximation error of these Spherical Random Fourier (SRF) features in Section 4, and study their performance on a variety of standard datasets including a large-scale experiment on ImageNet in Section 5 and in the Supplementary Material.

Compared to prior work, the SRF method is able to achieve lower kernel approximation error with compact nonlinear maps, especially for higher-order polynomials. The variance in kernel approximation error is much lower than that of existing techniques, leading to more stable predictions. In addition, it does not suffer from the rank deficiency problem seen in other methods. Before describing the SRF method in detail, we begin by reviewing the method of Random Fourier Features.

## 2 Background: Random Fourier Features

In [5], a method for the explicit construction of compact nonlinear randomized feature maps was presented. The technique relies on two important properties of the kernel: i) The kernel is shift-invariant, *i.e.* $K(\mathbf{x}, \mathbf{y}) = K(\mathbf{z})$ where $\mathbf{z} = \mathbf{x} - \mathbf{y}$ and ii) The function $K(\mathbf{z})$ is positive definite on $\mathbb{R}^d$. Property (ii) guarantees that the Fourier transform of $K(\mathbf{z})$, $k(\mathbf{w}) = \frac{1}{(2\pi)^{d/2}} \int d^d \mathbf{z} \, K(\mathbf{z}) \, e^{i \langle \mathbf{w}, \mathbf{z} \rangle}$, admits an interpretation as a probability distribution. This fact follows from Bochner's celebrated characterization of positive definite functions,

**Theorem 1.** *(Bochner [6]) A function $K \in C(\mathbb{R}^d)$ is positive definite on $\mathbb{R}^d$ if and only if it is the Fourier transform of a finite non-negative Borel measure on $\mathbb{R}^d$.*

A consequence of Bochner's theorem is that the inverse Fourier transform of $k(\mathbf{w})$ can be interpreted as the computation of an expectation, i.e.,

$$
\begin{aligned}
K(\mathbf{z}) &= \frac{1}{(2\pi)^{d/2}} \int d^d \mathbf{w} \, k(\mathbf{w}) \, e^{-i \langle \mathbf{w}, \mathbf{z} \rangle} \\
&= E_{\mathbf{w} \sim p(\mathbf{w})} \, e^{-i \langle \mathbf{w}, \mathbf{x} - \mathbf{y} \rangle} \\
&= 2 \, E_{\substack{\mathbf{w} \sim p(\mathbf{w}) \\ b \sim U(0, 2\pi)}} \left[ \cos(\langle \mathbf{w}, \mathbf{x} \rangle + b) \cos(\langle \mathbf{w}, \mathbf{y} \rangle + b) \right] ,
\end{aligned}
\tag{1}
$$

where $p(\mathbf{w}) = (2\pi)^{-d/2} k(\mathbf{w})$ and $U(0, 2\pi)$ is the uniform distribution on $[0, 2\pi)$. If the above expectation is approximated using Monte Carlo with $D$ random samples $\mathbf{w}_i$, then $K(\mathbf{x}, \mathbf{y}) \approx \langle Z(\mathbf{x}), Z(\mathbf{y}) \rangle$ with $Z(\mathbf{x}) = \sqrt{2/D} \left[ \cos(\mathbf{w}_1^T \mathbf{x} + b_1), ..., \cos(\mathbf{w}_D^T \mathbf{x} + b_D) \right]^T$. This identification is

made possible by property (i), which guarantees that the functional dependence on $\mathbf{x}$ and $\mathbf{y}$ factorizes multiplicatively in frequency space.

Such Random Fourier Features have been used to approximate different types of positive-definite shift-invariant kernels, including the Gaussian kernel, the Laplacian kernel, and the Cauchy kernel. However, they have not yet been applied to polynomial kernels, because this class of kernels does not satisfy the positive-definiteness prerequisite for the application of Bochner's theorem. This statement may seem counter-intuitive given the known result that polynomial kernels $K(\mathbf{x}, \mathbf{y})$ are positive definite *kernels*. The subtlety is that this statement does not necessarily imply that the associated single variable *functions* $K(\mathbf{z}) = K(\mathbf{x} - \mathbf{y})$ are positive definite on $\mathbb{R}^d$ for all $d$. We will prove this fact in the next section, along with the construction of an efficient and effective modification of the Random Fourier method that can be applied to polynomial kernels defined on the unit sphere.

## 3 Polynomial kernels on the unit sphere

In this section, we consider approximating the polynomial kernel defined on $\mathcal{S}^{d-1} \times \mathcal{S}^{d-1}$,

$$K(\mathbf{x}, \mathbf{y}) = \left(1 - \frac{||\mathbf{x} - \mathbf{y}||^2}{a^2}\right)^p = \alpha \left(q + \langle \mathbf{x}, \mathbf{y} \rangle\right)^p \tag{2}$$

with $q = a^2/2 - 1$, $\alpha = (2/a^2)^p$. We will restrict our attention to $p \geq 1, a \geq 2$.

The kernel is a shift-invariant radial function of the single variable $\mathbf{z} = \mathbf{x} - \mathbf{y}$, which with a slight abuse of notation we write as $K(\mathbf{x}, \mathbf{y}) = K(\mathbf{z}) = K(z)$, with $z = ||\mathbf{z}||$.[2] In Section 3.1, we show that the Fourier transform of $K(\mathbf{z})$ is not a non-negative function, so a straightforward application of Bochner's theorem to produce Random Fourier Features as in [5] is impossible in this case. Nevertheless, in Section 3.2, we propose a fast and accurate approximation of $K(\mathbf{z})$ by a surrogate positive definite function which enables us to construct compact Fourier features.

### 3.1 Obstructions to Random Fourier Features

Because $z = ||\mathbf{x} - \mathbf{y}|| = \sqrt{2 - 2\cos\theta} \leq 2$, the behavior of $K(z)$ for $z > 2$ is undefined and arbitrary since it does not affect the original kernel function in eqn. (2). On the other hand, it should be specified in order to perform the Fourier transform, which requires an integration over all values of $z$. We first consider the natural choice of $K(z) = 0$ for $z > 2$ before showing that all other choices lead to the same conclusion.

**Lemma 1.** *The Fourier transform of $\{K(\mathbf{z}), z \leq 2; 0, z > 2\}$ is not a non-negative function of $\mathbf{w}$ for any values of a, p, and d.*

*Proof.* (See the Supplementary Material for details). A direct calculation gives,

$$k(w) = \sum_{i=0}^{p} \frac{p!}{(p-i)!} \left(1 - \frac{4}{a^2}\right)^{p-i} \left(\frac{2}{a^2}\right)^i \left(\frac{2}{w}\right)^{d/2+i} J_{d/2+i}(2w),$$

where $J_\nu(z)$ is the Bessel function of the first kind. Expanding for large $w$ yields,

$$k(w) \sim \frac{1}{\sqrt{\pi w}} \left(1 - \frac{4}{a^2}\right)^p \left(\frac{2}{w}\right)^{d/2} \cos\left((d+1)\frac{\pi}{4} - 2w\right), \tag{3}$$

which takes negative values for some $w$ for all $a > 2$, $p$, and $d$. $\square$

So a Monte Carlo approximation of $K(\mathbf{z})$ as in eqn. (1) is impossible in this case. However, there is still the possibility of defining the behavior of $K(\mathbf{z})$ for $z > 2$ differently, and in such a way that the Fourier transform is positive and integrable on $\mathbb{R}^d$. The latter condition should hold for all $d$, since the vector dimensionality $d$ can vary arbitrarily depending on input data.

We now show that such a function cannot exist. To this end, we first recall a theorem due to Schoenberg regarding *completely monotone* functions,

**Definition 1.** *A function $f$ is said to be* completely monotone *on an interval $[a, b] \subset \mathbb{R}$ if it is continuous on the closed interval, $f \in C([a, b])$, infinitely differentiable in its interior, $f \subset C^\infty((a, b))$, and $(-1)^l f^{(l)}(x) \geq 0, \quad x \in (a, b), \ l = 0, 1, 2, \ldots$*

**Theorem 2.** *(Schoenberg [18]) A function $\phi$ is completely monotone on $[0, \infty)$ if and only if $\Phi \equiv \phi(|| \cdot ||^2)$ is positive definite and radial on $\mathbb{R}^d$ for all $d$.*

Together with Theorem 1, Theorem 2 shows that $\phi(z) = K(\sqrt{z})$ must be completely monotone if $k(\mathbf{w})$ is to be interpreted as a probability distribution. We now establish that $\phi(z)$ cannot be completely monotone and simultaneously satisfy $\phi(z) = K(\sqrt{z})$ for $z \leq 2$.

**Proposition 1.** *The function $\phi(z) = K(\sqrt{z})$ is completely monotone on $[0, a^2]$.*
*Proof.* From the definition of $\phi$, $\phi(z) = \left(1 - \frac{z}{a^2}\right)^p$, $\phi$ is continuous on $[0, a^2]$, infinitely differentiable on $(0, a^2)$, and its derivatives vanish for $l > p$. They obey $(-1)^l \phi^{(l)}(z) = \frac{p!}{(p-l)!} \frac{\phi(z)}{(a^2-z)^l} \geq 0$, where the inequality follows since $z < a^2$. Therefore $\phi$ is completely monotone on $[0, a^2]$. $\quad\square$

**Theorem 3.** *Suppose $f$ is a completely monotone polynomial of degree $n$ on the interval $[0, c]$, $c < \infty$, with $f(c) = 0$. Then there is no completely monotone function on $[0, \infty)$ that agrees with $f$ on $[0, a]$ for any nonzero $a < c$.*

*Proof.* Let $g \in C([0, \infty)) \bigcap C^\infty((0, \infty))$ be a non-negative function that agrees with $f$ on $[0, a]$ and let $h = g - f$. We show that for all non-negative integers $m$ there exists a point $\chi_m$ satisfying $a < \chi_m \leq c$ such that $h^{(m)}(\chi_m) > 0$. For $m = 0$, the point $\chi_0 = c$ obeys $h(\chi_0) = g(\chi_0) - f(\chi_0) = g(\chi_0) > 0$ by the definition of $g$. Now, suppose there is a point $\chi_m$ such that $a < \chi_m \leq c$ and $h^{(m)}(\chi_m) > 0$. The mean value theorem then guarantees the existence of a point $\chi_{m+1}$ such that $a < \chi_{m+1} < \chi_m$ and $h^{(m+1)}(\chi_{m+1}) = \frac{h^{(m)}(\chi_m) - h^{(m)}(a)}{\chi_m - a} = \frac{h^{(m)}(\chi_m)}{\chi_m - a} > 0$, where we have utilized the fact that $h^{(m)}(a) = 0$ and the induction hypothesis. Noting that $f^{(m)} = 0$ for all $m > n$, this result implies that $g^{(m)}(\chi_m) > 0$ for all $m > n$. Therefore $g$ cannot be completely monotone. $\quad\square$

**Corollary 1.** *There does not exist a finite non-negative Borel measure on $\mathbb{R}^d$ whose Fourier transform agrees with $K(\mathbf{z})$ on $[0, 2]$.*

### 3.2 Spherical Random Fourier features

From the section above, we see that the Bochner's theorem cannot be directly applied to the polynomial kernel. In addition, it is impossible to construct a positive integrable $\hat{k}(\mathbf{w})$ whose inverse Fourier transform $\hat{K}(\mathbf{z})$ equals $K(\mathbf{z})$ exactly on $[0, 2]$. Despite this result, it is nevertheless possible to find $\hat{K}(\mathbf{z})$ that is a good approximation of $K(\mathbf{z})$ on $[0, 2]$, which is all that is necessary given that we will be approximating $\hat{K}(\mathbf{z})$ by Monte Carlo integration anyway. We present our method of Spherical Random Fourier (SRF) features in this section.

We recall a characterization of radial functions that are positive definite on $\mathbb{R}^d$ for all $d$ due to Schoenberg.

**Theorem 4.** *(Schoenberg [18]) A continuous function $f : [0, \infty) \to \mathbb{R}$ is positive definite and radial on $\mathbb{R}^d$ for all $d$ if and only if it is of the form $f(r) = \int_0^\infty e^{-r^2 t^2} d\mu(t)$, where $\mu$ is a finite non-negative Borel measure on $[0, \infty)$.*

This characterization motivates an approximation for $K(z)$ as a sum of $N$ Gaussians, $\hat{K}(z) = \sum_{i=1}^N c_i e^{-\sigma_i^2 z^2}$. To increase the accuracy of the approximation, we allow the $c_i$ to take negative values. Doing so enables its Fourier transform (which is also a sum of Gaussians) to become negative. We circumvent this problem by mapping those negative values to zero,

$$\hat{k}(w) = \max\left(0, \ \sum_{i=1}^N c_i \left(\frac{1}{\sqrt{2}\sigma_i}\right)^d e^{-w^2/4\sigma_i^2}\right), \tag{4}$$

and simply defining $\hat{K}(z)$ as its inverse Fourier transform. Owing to the max in eqn. (4), it is not possible to calculate an analytical expression for $\hat{K}(z)$. Thankfully, this isn't necessary since we

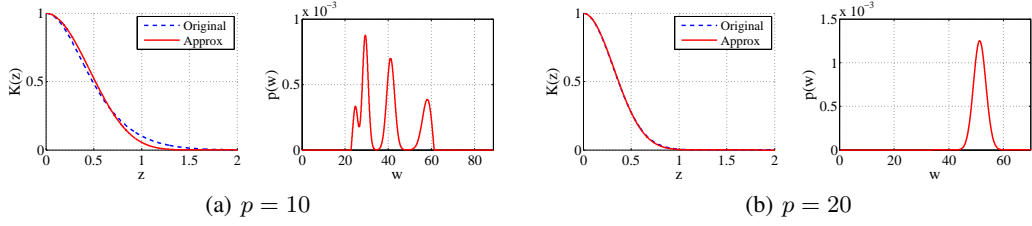

Figure 1: $K(z)$, its approximation $\hat{K}(z)$, and the corresponding pdf $p(w)$ for $d = 256$, $a = 2$ for polynomial orders (a) 10 and (b) 20. Higher-order polynomials are approximated better, see eqn. (6).

---

**Algorithm 1** Spherical Random Fourier (SRF) Features

---

**Input**: A polynomial kernel $K(\mathbf{x}, \mathbf{y}) = K(z)$, $z = ||\mathbf{x} - \mathbf{y}||_2$, $||\mathbf{x}||_2 = 1$, $||\mathbf{y}||_2 = 1$, with bias $a \geq 2$, order $p \geq 1$, input dimensionality $d$ and feature dimensionality D.
**Output**: A randomized feature map $Z(\cdot) : \mathbb{R}^d \to \mathbb{R}^D$ such that $\langle Z(\mathbf{x}), Z(\mathbf{y}) \rangle \approx K(\mathbf{x}, \mathbf{y})$.

1. Solve $\text{argmin}_{\hat{K}} \int_0^2 dz \left[ K(z) - \hat{K}(z) \right]^2$ for $\hat{k}(\mathbf{w})$, where $\hat{K}(\mathbf{z})$ is the inverse Fourier transform of $\hat{k}(\mathbf{w})$, whose form is given in eqn. (4). Let $p(\mathbf{w}) = (2\pi)^{-d/2} \hat{k}(\mathbf{w})$.
2. Draw $D$ *iid* samples $\mathbf{w}_1, ..., \mathbf{w}_D$ from $p(\mathbf{w})$.
3. Draw $D$ *iid* samples $b_1, ..., b_D \in \mathbb{R}$ from the uniform distribution on $[0, 2\pi]$.
4. $Z(\mathbf{x}) = \sqrt{\frac{2}{D}} \left[ \cos(\mathbf{w}_1^T \mathbf{x} + b_1), ..., \cos(\mathbf{w}_D^T \mathbf{x} + b_D) \right]^T$

---

can evaluate it numerically by performing a one dimensional numerical integral,

$$\hat{K}(z) = \int_0^\infty dw \, w \, \hat{k}(w)(w/z)^{d/2-1} J_{d/2-1}(wz) \,,$$

which is well-approximated using a fixed-width grid in $w$ and $z$, and can be computed via a single matrix multiplication. We then optimize the following cost function, which is just the MSE between $K(z)$ and our approximation of it,

$$L = \frac{1}{2} \int_0^2 dz \left[ K(z) - \hat{K}(z) \right]^2 \,, \tag{5}$$

which defines an optimal probability distribution $p(w)$ through eqn. (4) and the relation $p(\mathbf{w}) = (2\pi)^{-d/2} k(\mathbf{w})$. We can then follow the Random Fourier Feature [5] method to generate the nonlinear maps. The entire SRF process is summarized in Algorithm 1. Note that for any given of kernel parameters $(a, p, d)$, $p(\mathbf{w})$ can be pre-computed, independently of the data.

## 4 Approximation error

The total MSE comes from two sources: error approximating the function, *i.e.* $L$ from eqn. (5), and error from Monte Carlo sampling. The expected MSE of Monte-Carlo converges at a rate of $\mathcal{O}(1/D)$ and a bound on the supremum of the absolute error was given in [5]. Therefore, we focus on analyzing the first type of error.

We describe a simple method to obtain an upper bound on $L$. Consider the function $\hat{K}(z) = e^{-\frac{p}{a^2} z^2}$, which is a special case of eqn. (4) obtained by setting $N = 1$, $c_i = 1$, and $\sigma_1 = \sqrt{p/a^2}$. The MSE between $K(z)$ and this function thus provides an upper bound to our approximation error,

$$L = \frac{1}{2} \int_0^2 dz \, [\hat{K}(z) - K(z)]^2 \leq \frac{1}{2} \int_0^a dz \, [\hat{K}(z) - K(z)]^2$$

$$= \frac{1}{2} \int_0^a dz \left[ \exp\left(-\frac{2p}{a^2} z^2\right) + \left(1 - \frac{z^2}{a^2}\right)^{2p} - 2 \exp\left(-\frac{p}{a^2} z^2\right) \left(1 - \frac{z^2}{a^2}\right)^p \right]$$

$$= \frac{a}{4} \sqrt{\frac{\pi}{2p}} \, \text{erf}(\sqrt{2p}) + \frac{a}{4} \sqrt{\pi} \frac{\Gamma(p+1)}{\Gamma(p + \frac{3}{2})} - \frac{a}{2} \sqrt{\pi} \frac{\Gamma(p+1)}{\Gamma(p + \frac{3}{2})} M(\tfrac{1}{2}, p + \tfrac{3}{2}, -p) \,.$$

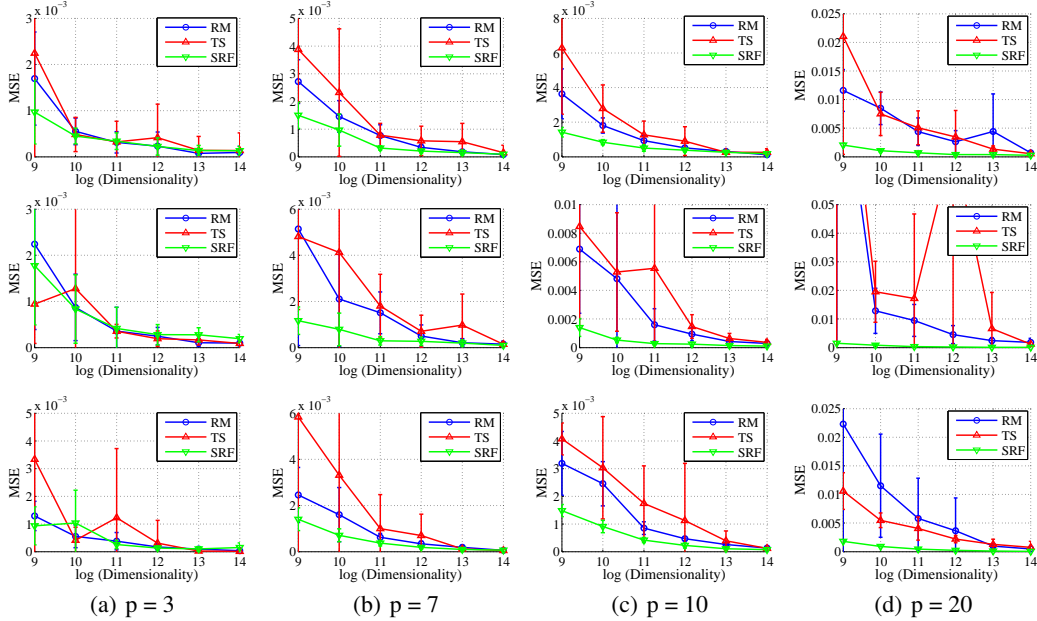

Figure 2: Comparison of MSE of kernel approximation on different datasets with various polynomial orders ($p$) and feature map dimensionalities. The first to third rows show results of `usps`, `gisette`, `adult`, respectively. SRF gives better kernel approximation, especially for large $p$.

In the first line we have used the fact that integrand is positive and $a \geq 2$. The three terms on the second line are integrated using the standard integral definitions of the error function, beta function, and Kummer's confluent hypergeometric function [19], respectively. To expose the functional dependence of this result more clearly, we perform an expansion for large $p$. We use the asymptotic expansions of the error function and the Gamma function,

$$\mathrm{erf}(z) = 1 - \frac{e^{-z^2}}{z\sqrt{\pi}} \sum_{k=0}^{\infty} (-1)^k \frac{(2k-1)!!}{(2z^2)^k} ,$$

$$\log \Gamma(z) = z \log z - z - \frac{1}{2} \log \frac{z}{2\pi} + \sum_{k=2}^{\infty} \frac{B_k}{k(k-1)} z^{1-k} ,$$

where $B_k$ are Bernoulli numbers. For the third term, we write the series representation of $M(a, b, z)$,

$$M(a, b, z) = \frac{\Gamma(b)}{\Gamma(a)} \sum_{k=0}^{\infty} \frac{\Gamma(a+k)}{\Gamma(b+k)} \frac{z^k}{k!} ,$$

expand each term for large $p$, and sum the result. All together, we obtain the following bound,

$$L \leq \frac{105}{4096} \sqrt{\frac{\pi}{2}} \frac{a}{p^{5/2}} , \tag{6}$$

which decays at a rate of $O(p^{-2.5})$ and becomes negligible for higher-order polynomials. This is remarkable, as the approximation error of previous methods increases as a function of $p$. Figure 1 shows two kernel functions $K(z)$, their approximations $\hat{K}(z)$, and the corresponding pdfs $p(w)$.

## 5 Experiments

We compare the SRF method with Random Maclaurin (RM) [14] and Tensor Sketch (TS) [15], the other polynomial kernel approximation approaches. Throughout the experiments, we choose the number of Gaussians, $N$, to equal 10, though the specific number had negligible effect on the results. The bias term is set as $a = 4$. Other choices such as $a = 2, 3$ yield similar performance; results with a variety of parameter settings can be found in the Supplementary Material. The error bars and standard deviations are obtained by conducting experiments 10 times across the entire dataset.

| Dataset | Method | $D = 2^9$ | $D = 2^{10}$ | $D = 2^{11}$ | $D = 2^{12}$ | $D = 2^{13}$ | $D = 2^{14}$ | Exact |
|---|---|---|---|---|---|---|---|---|
| usps $p = 3$ | RM | $87.29 \pm 0.87$ | $89.11 \pm 0.53$ | $90.43 \pm 0.49$ | $91.09 \pm 0.44$ | $91.48 \pm 0.31$ | $91.78 \pm 0.32$ | |
| | TS | $89.85 \pm 0.35$ | $90.99 \pm 0.42$ | $91.37 \pm 0.19$ | $91.68 \pm 0.19$ | $91.85 \pm 0.18$ | $91.90 \pm 0.23$ | $96.21$ |
| | SRF | $\mathbf{90.91 \pm 0.32}$ | $\mathbf{92.08 \pm 0.32}$ | $\mathbf{92.50 \pm 0.48}$ | $\mathbf{93.10 \pm 0.26}$ | $\mathbf{93.31 \pm 0.16}$ | $\mathbf{93.28 \pm 0.24}$ | |
| usps $p = 7$ | RM | $88.86 \pm 1.08$ | $91.01 \pm 0.44$ | $92.70 \pm 0.38$ | $94.03 \pm 0.30$ | $94.54 \pm 0.30$ | $94.97 \pm 0.26$ | |
| | TS | $92.30 \pm 0.52$ | $93.59 \pm 0.20$ | $94.53 \pm 0.20$ | $94.84 \pm 0.10$ | $95.06 \pm 0.23$ | $95.27 \pm 0.12$ | $96.51$ |
| | SRF | $\mathbf{92.44 \pm 0.31}$ | $\mathbf{93.85 \pm 0.32}$ | $\mathbf{94.79 \pm 0.19}$ | $\mathbf{95.06 \pm 0.21}$ | $\mathbf{95.37 \pm 0.12}$ | $\mathbf{95.51 \pm 0.17}$ | |
| usps $p = 10$ | RM | $88.95 \pm 0.60$ | $91.41 \pm 0.46$ | $93.27 \pm 0.28$ | $94.29 \pm 0.34$ | $95.19 \pm 0.21$ | $95.53 \pm 0.25$ | |
| | TS | $92.41 \pm 0.48$ | $93.85 \pm 0.34$ | $94.75 \pm 0.26$ | $95.31 \pm 0.28$ | $95.55 \pm 0.25$ | $\mathbf{95.91 \pm 0.17}$ | $96.56$ |
| | SRF | $\mathbf{92.63 \pm 0.46}$ | $\mathbf{94.33 \pm 0.33}$ | $\mathbf{95.18 \pm 0.26}$ | $\mathbf{95.60 \pm 0.27}$ | $\mathbf{95.78 \pm 0.23}$ | $95.85 \pm 0.16$ | |
| usps $p = 20$ | RM | $88.67 \pm 0.98$ | $91.09 \pm 0.42$ | $93.22 \pm 0.39$ | $94.32 \pm 0.27$ | $95.24 \pm 0.27$ | $95.62 \pm 0.24$ | |
| | TS | $91.73 \pm 0.88$ | $93.92 \pm 0.28$ | $94.68 \pm 0.28$ | $95.26 \pm 0.31$ | $95.90 \pm 0.20$ | $96.07 \pm 0.19$ | $96.81$ |
| | SRF | $\mathbf{92.27 \pm 0.48}$ | $\mathbf{94.30 \pm 0.46}$ | $\mathbf{95.48 \pm 0.39}$ | $\mathbf{95.97 \pm 0.32}$ | $\mathbf{96.18 \pm 0.23}$ | $\mathbf{96.28 \pm 0.15}$ | |
| gisette $p = 3$ | RM | $89.53 \pm 1.43$ | $92.77 \pm 0.40$ | $94.49 \pm 0.48$ | $95.90 \pm 0.31$ | $96.69 \pm 0.33$ | $97.01 \pm 0.26$ | |
| | TS | $\mathbf{93.52 \pm 0.60}$ | $\mathbf{95.28 \pm 0.71}$ | $\mathbf{96.12 \pm 0.36}$ | $\mathbf{96.76 \pm 0.40}$ | $\mathbf{97.06 \pm 0.19}$ | $\mathbf{97.12 \pm 0.27}$ | $98.00$ |
| | SRF | $91.72 \pm 0.92$ | $94.39 \pm 0.65$ | $95.62 \pm 0.47$ | $96.50 \pm 0.40$ | $96.91 \pm 0.36$ | $97.05 \pm 0.19$ | |
| gisette $p = 7$ | RM | $89.44 \pm 1.44$ | $92.77 \pm 0.57$ | $95.15 \pm 0.60$ | $96.37 \pm 0.46$ | $96.90 \pm 0.46$ | $97.27 \pm 0.22$ | |
| | TS | $\mathbf{92.89 \pm 0.66}$ | $\mathbf{95.29 \pm 0.39}$ | $96.32 \pm 0.47$ | $96.66 \pm 0.34$ | $97.16 \pm 0.25$ | $\mathbf{97.58 \pm 0.25}$ | $97.90$ |
| | SRF | $92.75 \pm 1.01$ | $94.85 \pm 0.53$ | $\mathbf{96.42 \pm 0.49}$ | $\mathbf{97.07 \pm 0.30}$ | $\mathbf{97.50 \pm 0.24}$ | $97.53 \pm 0.15$ | |
| gisette $p = 10$ | RM | $89.91 \pm 0.58$ | $93.16 \pm 0.40$ | $94.94 \pm 0.72$ | $96.19 \pm 0.49$ | $96.88 \pm 0.23$ | $97.15 \pm 0.40$ | |
| | TS | $\mathbf{92.48 \pm 0.62}$ | $94.61 \pm 0.60$ | $95.72 \pm 0.53$ | $96.60 \pm 0.58$ | $96.99 \pm 0.28$ | $97.41 \pm 0.20$ | $98.10$ |
| | SRF | $92.42 \pm 0.85$ | $\mathbf{95.10 \pm 0.47}$ | $\mathbf{96.35 \pm 0.42}$ | $\mathbf{97.15 \pm 0.34}$ | $\mathbf{97.57 \pm 0.23}$ | $\mathbf{97.75 \pm 0.14}$ | |
| gisette $p = 20$ | RM | $89.40 \pm 0.98$ | $92.46 \pm 0.67$ | $94.37 \pm 0.55$ | $95.67 \pm 0.43$ | $96.14 \pm 0.55$ | $96.63 \pm 0.40$ | |
| | TS | $90.49 \pm 1.07$ | $92.88 \pm 0.42$ | $94.43 \pm 0.69$ | $95.41 \pm 0.71$ | $96.24 \pm 0.44$ | $96.97 \pm 0.28$ | $98.00$ |
| | SRF | $\mathbf{92.12 \pm 0.62}$ | $\mathbf{94.22 \pm 0.45}$ | $\mathbf{95.85 \pm 0.54}$ | $\mathbf{96.94 \pm 0.29}$ | $\mathbf{97.47 \pm 0.24}$ | $\mathbf{97.75 \pm 0.32}$ | |

Table 1: Comparison of classification accuracy (in %) on different datasets for different polynomial orders ($p$) and varying feature map dimensionality (D). The *Exact* column refers to the accuracy of exact polynomial kernel trained with libSVM. More results are given in the Supplementary Material.

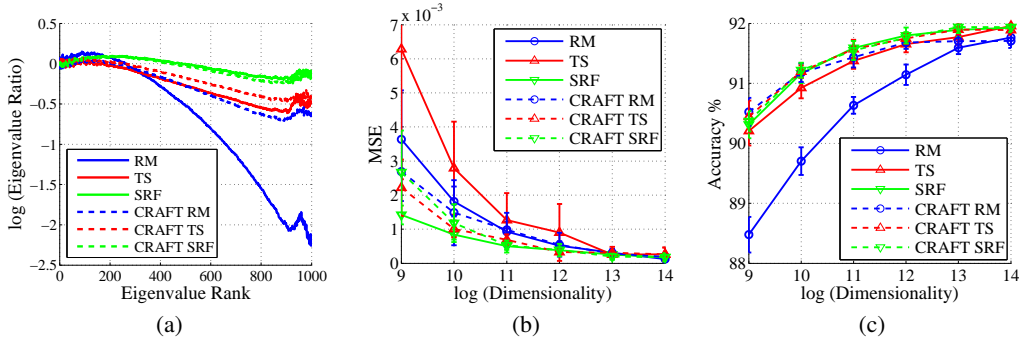

(a)  (b)  (c)

Figure 3: Comparison of CRAFT features on usps dataset with polynomial order $p = 10$ and feature maps of dimension $D = 2^{12}$. (a) Logarithm of ratio of $i$th-leading eigenvalue of the approximate kernel to that of the exact kernel, constructed using 1,000 points. CRAFT features are projected from $2^{14}$ dimensional maps. (b) Mean squared error. (c) Classification accuracy.

**Kernel approximation.** The main focus of this work is to improve the quality of kernel approximation, which we measure by computing the mean squared error (MSE) between the exact kernel and its approximation across the entire dataset. Figure 2 shows MSE as a function of the dimensionality (D) of the nonlinear maps. SRF provides lower MSE than other methods, especially for higher order polynomials. This observation is consistent with our theoretical analysis in Section 4. As a corollary, SRF provides more compact maps with the same kernel approximation error. Furthermore, SRF is stable in terms of the MSE, whereas TS and RM have relatively large variance.

**Classification with linear SVM.** We train linear classifiers with liblinear [3] and evaluate classification accuracy on various datasets, two of which are summarized in Table 1; additional results are available in the Supplementary Material. As expected, accuracy improves with higher-dimensional nonlinear maps and higher-order polynomials. It is important to note that better kernel approximation does not necessarily lead to better classification performance because the original kernel might not be optimal for the task [20, 21]. Nevertheless, we observe that SRF features tend to yield better classification performance in most cases.

**Rank-Deficiency.** Hamid et al. [16] observe that RM and TS produce nonlinear features that are rank deficient. Their approximation quality can be improved by first mapping the input to a higher dimensional feature space, and then randomly projecting it to a lower dimensional space. This method is known as CRAFT. Figure 3(a) shows the logarithm of the ratio of the $i$th eigenvalue

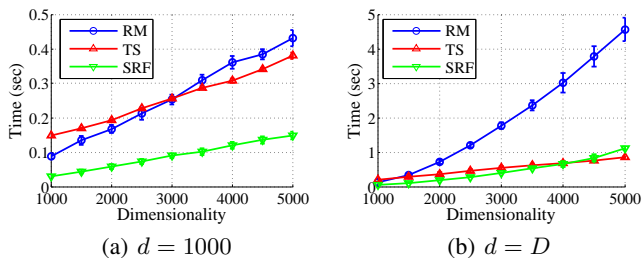
(a) $d = 1000$          (b) $d = D$

Figure 4: Computational time to generate randomized feature map for 1,000 random samples on a fixed hardware with $p = 3$. (a) $d = 1,000$. (b) $d = D$.

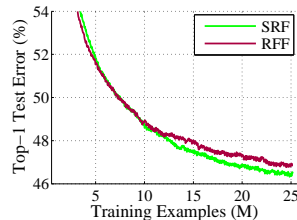

Figure 5: Doubly stochastic gradient learning curves with RFF and SRF features on ImageNet.

of the various approximate kernel matrices to that of the exact kernel. For a full-rank, accurate approximation, this value should be constant and equal to zero, which is close to the case for SRF. RM and TS deviate from zero significantly, demonstrating their rank-deficiency.

Figures 3(b) and 3(c) show the effect of the CRAFT method on MSE and classification accuracy. CRAFT improves RM and TS but it has no or even a negative effect on SRF. These observations all indicate that the SRF is less rank-deficient than RM and TS.

**Computational Efficiency.** Both RM and SRF have computational complexity $\mathcal{O}(ndD)$, whereas TS scales as $\mathcal{O}(np(d + D \log D))$, where $D$ is the number of nonlinear maps, $n$ is the number of samples, $d$ is the original feature dimension, and $p$ is the polynomial order. Therefore the scalability of TS is better than SRF when $D$ is of the same order as $d$ ($\mathcal{O}(D \log D)$ vs. $\mathcal{O}(D^2)$). However, the computational cost of SRF does not depend on $p$, making SRF more efficient for higher-order polynomials. Moreover, there is little computational overhead involved in the SRF method, which enables it to outperform $TS$ for practical values of $D$, even though it is asymptotically inferior. As shown in Figure 4(a), even for the low-order case ($p = 3$), SRF is more efficient than TS for a fixed $d = 1000$. In Figure 4(b), where $d = D$, SRF is still more efficient than TS up to $D \lesssim 4000$.

**Large-scale Learning.** We investigate the scalability of the SRF method on the ImageNet 2012 dataset, which consists of 1.3 million $256 \times 256$ color images from 1000 classes. We employ the doubly stochastic gradient method of Dai et al. [22], which utilizes two stochastic approximations — one from random training points and the other from random features associated with the kernel. We use the same architecture and parameter settings as [22] (including the fixed convolutional neural network parameters), except we replace the RFF kernel layer with an $\ell_2$ normalization step and an SRF kernel layer with parameters $a = 4$ and $p = 10$. The learning curves in Figure 5 suggest that SRF features may perform better than RFF features on this large-scale dataset. We also evaluate the model with multi-view testing, in which max-voting is performed on 10 transformations of the test set. We obtain Top-1 test error of $44.4\%$, which is comparable to the $44.5\%$ error reported in [22]. These results demonstrate that the unit norm restriction does not have a negative impact on performance in this case, and that polynomial kernels can be successfully scaled to large datasets using the SRF method.

# 6   Conclusion

We have described a novel technique to generate compact nonlinear features for polynomial kernels applied to data on the unit sphere. It approximates the Fourier transform of kernel functions as the positive projection of an indefinite combination of Gaussians. It achieves more compact maps compared to the previous approaches, especially for higher-order polynomials. SRF also shows less feature redundancy, leading to lower kernel approximation error. Performance of SRF is also more stable than the previous approaches due to reduced variance. Moreover, the proposed approach could easily extend beyond polynomial kernels: the same techniques would apply equally well to any shift-invariant radial kernel function, positive definite or not. In the future, we would also like to explore adaptive sampling procedures tuned to the training data distribution in order to further improve the kernel approximation accuracy, especially when $D$ is large, *i.e.* when the Monte-Carlo error is low and the kernel approximation error dominates.

**Acknowledgments.** We thank the anonymous reviewers for their valuable feedback and Bo Xie for facilitating experiments with the doubly stochastic gradient method.

## Footnotes

[1]We are not claiming total generality of this setting; nevertheless, in cases where the vector length carries useful information and should be preserved, it could be added as an additional feature before normalization.

[2]We also follow this practice in frequency space, i.e. if $k(\mathbf{w})$ is radial, we also write $k(\mathbf{w}) = k(w)$.

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
