[Supplementary Material · poly_supp.pdf]

# Spherical Random Features for Polynomial Kernels
## Supplementary Material

## 1 Proof of lemma

**Lemma 1.** *The Fourier transform of $K(\mathbf{z}) = \{(1 - ||\mathbf{z}||^2/a^2)^p, z \leq 2; 0, z > 2\}$ is not a non-negative function of $\mathbf{w}$ for any values of $a$, $p$, and $d$.*

*Proof.* The Fourier transformation of $K(\mathbf{z})$ can be simplified using spherical coordinates as,

$$
\begin{aligned}
k(\mathbf{w}) &= \frac{1}{(2\pi)^{d/2}} \int d^d\mathbf{z}\, K(\mathbf{z})\, e^{i\langle \mathbf{w}, \mathbf{z}\rangle} \\
&= \frac{\text{Vol}(S^{d-2})}{(2\pi)^{d/2}} \int_0^2 dz\, z^{d-1}\, K(z) \int_0^\pi d\theta \sin^{d-2}\theta\, e^{izw\cos\theta} \\
&= \int_0^2 dz\, z\, K(z)(z/w)^{d/2-1} J_{d/2-1}(zw)
\end{aligned}
$$

where $w = ||\mathbf{w}||$, $z = ||\mathbf{z}||$ and $J_\nu(x)$ is the Bessel function of the first kind. Because the kernel is a radial function of $\mathbf{z}$, *i.e.* $K(\mathbf{z}) = K(z)$, it is also a radial function of $\mathbf{w}$, *i.e.* $k(\mathbf{w}) = k(w)$.

For the final integration, we use a binomial expansion to rewrite $K(z)$ in powers of $(1 - z^2/4)$,

$$
K(z) = \left(1 - \frac{z^2}{a^2}\right)^p = \sum_{i=0}^p \binom{p}{i} \left(1 - \frac{4}{a^2}\right)^{p-i} \left(\frac{4}{a^2}\right)^i \left(1 - \frac{z^2}{4}\right)^i.
$$

This expression is convenient because each of term can be integrated in terms of Bessel functions,

$$
k(w) = \sum_{i=0}^p \frac{p!}{(p-i)!} \left(1 - \frac{4}{a^2}\right)^{p-i} \left(\frac{2}{a^2}\right)^i \left(\frac{2}{w}\right)^{d/2+i} J_{d/2+i}(2w),
$$

which holds even for $a = 2$ if we take the limit $a \to 2$ after performing the sum. Owing to the oscillatory behavior of the Bessel functions, $k(w)$ is not a non-negative function. This can be seen explicitly by examining the asymptotic behavior for large $w$ ($w \to \infty$), which is dominated by the $i = 0$ term in the sum,

$$
k(w) \sim \frac{1}{\sqrt{\pi w}} \left(1 - \frac{4}{a^2}\right)^p \left(\frac{2}{w}\right)^{d/2} \cos\left((d+1)\frac{\pi}{4} - 2w\right),
$$

which takes negative values for some $w$ for all $a \neq 2$, $p$, and $d$. For the case $a = 2$, the $i = 0$ term is absent, but the same result holds if we take $p \to 0$. $\qquad\square$

## 2 Datasets

| Dataset | Num. Training | Num. Testing | Dim. |
|---------|---------------|--------------|------|
| usps    | 7,291         | 2,007        | 256  |
| gisette | 6,000         | 1,000        | 5,000 |
| adult   | 32,561        | 16,281       | 123  |
| mnist   | 60,000        | 10,000       | 784  |

Table 1: Datasets used in the experiments.

## 3 Kernel approximation and associated probability distributions

We show the approximation of $K(z)$ under different parameters in Figure 1 - 6. We fixed the number of Gaussians to be 10, but the specific choice has little effect on performance. Although the max in eqn. (4) is not differentiable, we find that when optimization is performed with L-BFGS, it usually converges within tens of iterations and is relatively stable with respect to random initializations.

Figure 1: $d = 256$, $a = 2$. The first row shows $K(z)$ and its approximation $\hat{K}(z)$. The second row shows the pdf of the approximation $p(w)$.

Figure 2: $d = 256$, $a = 3$. The first row shows $K(z)$ and its approximation $\hat{K}(z)$. The second row shows the pdf of the approximation $p(w)$.

Figure 3: $d = 256$, $a = 4$. The first row shows $K(z)$ and its approximation $\hat{K}(z)$. The second row shows the pdf of the approximation $p(w)$.

Figure 4: $d = 5000$, $a = 2$, The first row shows $K(z)$ and its approximation $\hat{K}(z)$. The second row shows the pdf of the approximation $p(w)$.

Figure 5: $d = 5000$, $a = 3$. The first row shows $K(z)$ and its approximation $\hat{K}(z)$. The second row shows the pdf of the approximation $p(w)$.

Figure 6: $d = 5000$, $a = 4$. The first row shows $K(z)$ and its approximation $\hat{K}(z)$. The second row shows the pdf of the approximation $p(w)$.

# 4 Additional experiments on MSE and classification accuracy

We show additional experiment results in Figure 7 - 17. The $C$ parameter of SVM is tuned on the scale of $\{0.1, 1, 10\}$. The bias term $a$ is varied across the experiments, but it does have a major influence on the kernel approximation or classification performance.

Figure 7: usps a = 2. The first row shows the MSE. The second row shows the classification accuracy.

Figure 8: usps a = 3. The first row shows the MSE. The second row shows the classification accuracy.

Figure 9: `usps` a = 4. The first row shows the MSE. The second row shows the classification accuracy.

Figure 10: `gisette` a = 2. The first row shows the MSE. The second row shows the classification accuracy.

Figure 11: `gisette` a = 3. The first row shows the MSE. The second row shows the classification accuracy.

Figure 12: `gisette` a = 4. The first row shows the MSE. The second row shows the classification accuracy.

Figure 13: `adult` a = 2. The first row shows the MSE. The second row shows the classification accuracy.

Figure 14: `adult` a = 3. The first row shows the MSE. The second row shows the classification accuracy.

Figure 15: `adult` a = 4. The first row shows the MSE. The second row shows the classification accuracy.

Figure 16: `mnist` a = 3. The first row shows the MSE. The second row shows the classification accuracy.

Figure 17: `mnist` a = 4. The first row shows the MSE. The second row shows the classification accuracy.