[Reviews · NeurIPS 2015]

Submitted by Assigned_Reviewer_1

== Summary ==

This paper proposes a Spherical Random Fourier (SRF) features for polynomial kernels for data on the unit sphere. Although the polynomial kernels on the unit sphere is known to be shift-invariant, the paper shows that it is not always a positive definite function which prevents one from applying Bochner's theorem and obtain random Fourier features. Based on Schoenberg's characterization of positive definite function on an interval [0,2], they propose to approximate the kernel as an indefinite sum of Gaussians. The optimal probability p(w) from which the random features are sampled is obtained by minimizing the MSE between the true kernel K(z) and its approximation. Compared to Random Maclaurin (RM) and Tensor Sketch (TS), the proposed technique seems to enjoy better approximation as well as predictive accuracy, especially for polynomial of large degree p. In many datasets, however, it seems to perform as good as TS in term of accuracy, e.g., USPS. It's very nice to also see the extensive experimental results in the supplementary.

In summary, this is a nice paper, although I see some disconnection between motivation in using Schoenberg's result (Theorem 4). However, I believe Schoenberg's characterization of kernel can be applied more generally. The empirical results are encouraging.

== Quality ==

The paper is technically sound. The major criticism I have is on Section 3.2 and Section 4.

Regarding Section 3.2, it is not clear why Schoenberg's characterization is a suitable way of approximating the K(z) on [0,2]. It looks like one can construct an approximation of K(z) directly on [0,2] using arbitrary functions. Better justification should be provided.

In Section 4, the approximation error decays at a rate of O(p^{-2.5}) which is a bit obscure as the analysis is done with fixed N=1. How does the number of Gaussians, N, affect the approximation? In experiment, it is mentioned that the specific number of N had negligible effect on the result. Why is this the case? Can you incorporate N into the analysis in Section 4 so it becomes clearer?

Some comments/questions: [Line 086-089] To interpret the Fourier transform as a probability distribution, should one also assume that K(0) = 1?

== Clarity ==

The paper is clearly written. I only have minor comments:

The notation d^d in Eq. (1) is confusing. There seems to be a disconnection between the first part and the approximation of the kernel.

Comments/questions: [Line 298] what does (2k - 1)!! mean mathematically?

== Originality ==

Previous works in large-scale kernel learning oftens employed Bochner's characterization of positive definite kernel. On the other hand, this paper took another path and used the characterization of Schoenberg which is less known in the literature. One can argue, however, that using Schoenberg's characterization may not be the best option.

== Significance ==

It looks like the Schoenberg's characterization (Theorem 4) can be applied more generally, not just to polynomial kernels.

One drawback of this paper is the restriction to polynomial kernel although the Schoenberg's characterization can already be applied to more general class of shift-invariant kernels. It is not clear why this characterization is particularly useful for polynomial kernels. What is the main difficulty?

Minor comments

In many datasets, the performance of SRF is comparable to that of TS. What is the intuition?
Summary: An interesting random features formulation of polynomial kernels for large-scale learning. Accept.

Submitted by Assigned_Reviewer_2

Overall a very good paper which provides a lot of insight onto the case of normalized data and random features for polynomial kernel.

Every step is very well motivated and the contribution is clear and novel.

Experiments are rigorous and shows clear advantage.
Summary: The paper proposes a kernel feature approximation for polynomial kernels when the norms of the element are constant. It is know that utilizing norm information improves the estimates of Random Projections

(please cite Li et. al. "Improving Random Projections Using Marginal Information" COLT 2006) and so this is indeed good directions to pursue and imporvement are expected.

The authors shows a complete results that the monte carlo approximation of Rahimi and Recht is not possible and this motivates them to go for approximations.

The form of approximation as linear combination of Gaussian is well motivated via Schoenbergs theorem. The authors further provide approximation gurantees which adds to the contribution.

Experiments are thorough and

compares with the two best baselines in the literature random mclauren and tensor sketch on both approximation accuracy and classification task.

Submitted by Assigned_Reviewer_3

The paper provides a technique for approximating polynomial kernels when the data are on the

unit sphere. The authors show that while in this case the kernel seems to be shift invariant Bochner's Theorem does not apply (because the kernel has to be shift invariant in the whole domain) and the Fourier transform contains negative components. Nevertheless, it is still

possible to approximate it with the non-negative part of linear combination of gaussians. The authors provide plots of the sampling distributions they obtain when they minimize the

approximation error between the kernel induced by the sum of gaussians and the actual kernel, bounds on the approximation error and empirical results showing that the proposed technique

mostly outperforms previous techniques such as Tensor Sketch and Random Mclaurin series.

In general the paper is well written and the idea of approximating kernels with some negative components in the Fourier domain with non-negative functions is original. However, I'd like

to see the following items addressed: -The authors did not explain exactly which parameters are being optimized by their LBFGS

procedure. Is it just the c_i or the c_i as well as the variances sigma_i^2?

-Also, it is not clear how the sampling from p(w) is performed. As shown in the plots, p(w) can be

multimodal and non trivial to sample from. It would be great if a description was provided. -Finally, do the timings reported include the setup time (i.e. LBFGS and sampling)?

AFTER REBUTTAL: The authors have answered the above questions in a satisfactory way. LBFGS optimizes all parameters, sampling is via inverse CDF, timings do not include LBFGS and sampling but those times are small compared to training.
Summary: The paper proposes a novel way to construct randomized approximations to polynomial kernels, which seems to work well empirically even though it introduces two levels of approximation (monte carlo sampling and a projection). The paper is clearly written and the experiments are carefully designed, comparing with many existing techniques on a variety of datasets.

Submitted by Assigned_Reviewer_4

The paper proposes to generate the random features for the polynomial kernel. The observation is that Bochner's theorem cannot be directly applied for polynomial kernel, and thus for normalized data, they propose to use combination of Gaussians to approximate the Fourier transform in random feature.

I think the paper is well motivated and easy to read. Still I have some questions about the paper:

1: Throughout the paper, I am not quite sure why the proposed method can perform better than other random feature based polynomial kernel approximation methods, e.g.,.[16][14]. The proposed method even has very strict assumption of the data (all the data should be on the unit sphere), so I think it would be better to highlight the difference/advantage of the proposed method comparing with other polynomial kernel approximation methods.

2: It would be interesting to show the time cost for each step of the proposed method. For example, time for computing k(w), and generate features.

3: How is the proposed method compared with other polynomial kernel approximation methods other than random features? for example, Nystrom approximation can also been used for approximating polynomial kernels.
Summary: The paper proposes to generate the random features for the polynomial kernel. The observation is that Bochner's theorem cannot be directly applied for polynomial kernel, and thus for normalized data, they propose to use combination of Gaussians to approximate the Fourier transform in random feature.

Author Feedback
Author rebuttal: We thank the reviewers for their comments. Below is our response to specific questions:

R1: "It looks like one can construct an approximation of K(z) directly on [0,2] using arbitrary functions"
As discussed in Section 3.1, complete monotonicity imposes strong constraints on the approximation for z>2. We choose Schoenberg's characterization for the basis of approximation functions since it manifestly obeys these constraints. Of course other bases could be used, so long as the requisite behavior for z>2 is obeyed. We don't anticipate the choice of basis to have a strong effect on the results.

R1: "Schoenberg's characterization can be applied more generally, not just to polynomial kernels."
Indeed, the proposed method could be applied to any shift-invariant kernel. We chose to focus on the polynomial case for concreteness and due to its practical utility. In the final version we will further emphasize the method's greater generality.

R1, R3: "How does the number of Gaussians, N, affect the approximation", "Limitations of the approximation analysis/bound"
We provide an upper bound on the approximation error by examining an explicit single-Gaussian approximation. Of course, using more Gaussians cannot make this approximation worse. This approach allows us to obtain a simple analytical expression which provides useful intuition for how the bound depends on the relevant parameters, particularly p.

The required number of Gaussians is data-dependent. We examine the approximation accuracy across a variety of datasets in Fig. 2 and throughout the supplementary material. This analysis provides a good sense of how well the approximations work in practice. We observe that a single Gaussian is often sufficient for large p (p=20), whereas roughly 4 Gaussians are needed for p=10.

R2: "Which parameters are being optimized by their L-BFGS procedure"
Both the variance and the c_i.

R2: "It is not clear how the sampling from p(w) is performed."
Since p(w) is a 1D function, the sampling can be simply carried out by inverse CDF sampling.

R2, R3: "Offline computational time"
We have provided a detailed computational efficiency analysis in lines 397-404. Note that the computational time is the online nonlinear mapping time. The offline L-BFGS step is very efficient, as stated in the supplementary material, "it usually converges within tens of iterations and is relatively stable with respect to random initializations". More importantly, for any given p and d, the approximation of p(w) can be precomputed, independently of the data. We will clarify this in the final version.

R3: "Why does the proposed method perform better than other random feature methods [16][14]"? "The proposed method has restricted assumption ... on unit sphere".
To achieve low error for higher order polynomials, refs. [14,16] need higher-order monomials and higher-order tensor products. The quality of the approximation therefore deteriorates for large p. In contrast, the approximation error in our method decreases at a rate of O(p^{-2.5}) (Sec 4), leading to compact features compared to refs. [14,16], especially for high-order polynomials. It is correct that the methods in refs. [14,16] are slightly more general (lines 59-62) but the question we ask in this paper is can one have better approximation of polynomial kernels in the most widely used setting, i.e., when data is normalized to unit norm (e.g., Hamid et al., 2014). In an uncommon case, if data norm carries useful information, it can be added as an additional feature.

R3: "Other related works, e.g., Nystrom"
We agree with R3 there are other alternatives for kernel approximation. For example, the Nystrom method is perhaps as popular as nonlinear maps. The connections between Nystrom and nonlinear maps have been extensively studied before. For example, for the Nystrom approximations to work well, the eigenspectrum of the kernel matrix should have a large gap [*]. We would provide more comments in the final version.
[*] Yang et al. Nystrom method vs random fourier features: A theoretical and empirical comparison. NIPS 2012

R1: "What does (2k - 1)!! mean mathematically?"
It is the standard notation for double factorial.

R6: "It is known that utilizing norm information improves the estimates of Random Projections"
We thank the reviewer for the reference, and we will cite the COLT paper in our final version.